# Testing a 5G Communication System: Kriging-Aided O2I Path Loss Modeling Based on 3.5 GHz Measurement Analysis

**DOI:** 10.3390/s21206716

**Published:** 2021-10-09

**Authors:** Melissa Eugenia Diago-Mosquera, Alejandro Aragón-Zavala, Mauricio Rodriguez

**Affiliations:** 1Escuela de Ingeniería y Ciencias, Tecnologico de Monterrey, Av. Epigmenio González 500, Fracc. San Pablo, Querétaro 76130, Mexico; aaragon@tec.mx; 2Escuela de Ingeniería Eléctrica, Pontificia Universidad Católica de Valparaíso, Valparaíso 2362804, Chile; mauricio.rodriguez.g@pucv.cl

**Keywords:** Kriging, outdoor-to-indoor (O2I), path loss, radio propagation, shadowing

## Abstract

Deep knowledge of how radio waves behave in a practical wireless channel is required for the effective planning and deployment of radio access networks in outdoor-to-indoor (O2I) environments. Using more than 400 non-line-of-sight (NLOS) radio measurements at 3.5 GHz, this study analyzes and validates a novel O2I measurement-based path loss prediction narrowband model that characterizes and estimates shadowing through Kriging techniques. The prediction results of the developed model are compared with those of the most traditional assumption of slow fading as a random variable: COST231, WINNER+, ITU-R, 3GPP urban microcell O2I models and field measured data. The results showed and guaranteed that the predicted path loss accuracy, expressed in terms of the mean error, standard deviation and root mean square error (RMSE) was significantly better with the proposed model; it considerably decreased the average error for both scenarios under evaluation.

## 1. Introduction

In recent years, there is a clear need to supply a sufficiently high data rate for areas with elevated user volume such as venues, hotels, and conference centers, etc., where a lack of signal is evident. Usually, in these situations the deployment of a new indoor radio cell is unnecessary considering outdoor radio cell availability. Under these conditions, outdoor-to-indoor (O2I) models become very relevant. These models characterize signal propagation inside buildings coming from external mobile radio base stations (BSs), which are mounted across a network of outdoor sites, occupying towers on hilltops, rooftops in built-up areas, and other promising outdoor structures. O2I radio propagation has become a challenging work, according to Small Cell Forum [1] some reasons for this are:Due to the lower operating frequencies becoming exhausted, higher frequencies are being deployed. However, they are not as effective for range and building penetration.O2I signal propagation is affected because the building fabric is more eco-friendly and noise-free, using low-emissivity glass which reflects the radiation from cellular antennas, and soundproof materials which attenuate radio waves.There are large increases in demand for mobile cellular services which densify available networks with more sites and an increase in the size of the spectrum that mobile network operators (MNOs) can deploy on those sites. This produces localized congestion; the service may be available, but it may not always be satisfactory.

Therefore, the scientific community is encouraged to understand O2I radio wave propagation [2,3,4,5] to help radio network engineers to achieve efficient radio coverage estimation, determine the optimal BS location, and perform interference feasibility studies. In seeking to understand those links, the applicability of standard urban microcell O2I path loss models such as COST231, WINNER+, ITU-R, and 3GPP are empirically tested [6].

The COST 231 project is based on adjusted models such as the one-slope model (1SM), the multi-wall model (MWM) and the linear attenuation model (LAM), that are based on propagation measurements [7]. The final O2I model proposed by the COST231 project is based on empirical data acquired for NLOS links, where the model relates the loss inside a room to the loss measured outside of it on the side nearest to the wall of interest, i.e., multi-floor propagation. The WINNER+ channel model followed a geometry-based stochastic channel modeling approach [8]. The channel parameters were determined stochastically, based on statistical distributions extracted from channel measurements. The ITU-R model [9] provided guidelines for both the procedure and the criteria (technical, spectrum and service) to be used in evaluating the proposed IMT-Advanced radio interface technologies (RITs) or sets of RITs (SRITs) for a number of test environments and deployment scenarios for the evaluation of the band of frequencies between 2 and 6 GHz. The NLOS model proposed by the 3GPP is based on measurement campaigns, for O2I building penetration loss. This model considered the material penetration losses for two types of models: low loss and high loss. The composition of low and high loss is dependent on the use of metal-coated glass in buildings and the deployment scenario characteristics. The study reported for the 3GPP project considered not only O2I building penetration losses but also O2I car penetration losses for the frequency range of 0.6 to 60 GHz [10].

According to [11], through an accurate channel modeling: the in-building radio propagation phenomena, complex by nature, can be characterized; the range of a wireless communication system can be calculated by assessing the expected coverage inside a building; signal strength/path loss can be predicted more accurately everywhere in a building; and channel performance predictions can be made quickly.

There are essentially three approaches for propagation models: physical, empirical and hybrid. The majority of physical models are simple to use but their assumptions are based on many simplifications. Generally, for this reason, they are employed to describe the phenomenon within a given error, whereas empirical models take valuable and building-specific information into account. As a result of combining the previous approaches, hybrid models include the accuracy of physical models and the suitability of measurement-based tuning perform in empirical models. On this basis, measurement-based methods are promising to achieve accurate and practical predictions, even in situations where there are not enough samples to carry out a rigorous characterization. To address the shortcoming of samples from measurement campaigns, linear geostatistics demonstrate their usefulness to predict unknown data with few available samples for practical designs.

To quantify the reliability of coverage provided by any radio cell it is imperative to understand and to characterize median path loss, shadow fading, and fast fading as the main path loss components [12]. As is surveyed in [11], the estimation of median path loss is deterministic, mathematical models describe it in terms of how path loss changes as a function of some factors, such as frequency and specific distance, etc. Nevertheless, these models need to account for the shadowing process, including it as an additional variable which describes the dispersion with respect to the nominal value given by the path loss models. Therefore, due to the normal distribution that shadowing produces in the signal measured, the most traditional action is to characterize it by a zero-mean Gaussian random variable, such as in [6]. Other wireless studies include nonconstant shadowing variance and non-lognormal shadowing, and predict the variance by considering the correlation between paths [13,14]. However, in [15] the authors validate a more accurate method to estimate the spatial correlation of shadowing by including Kriging, a linear geostatistics technique that is based on the regionalized theory, which states that there is a variance rate between samples over space in a physical continuity context, i.e., the spatial dependence stated by Tobler in [16]: spatial samples taken close to each other may be expected to have more similar values than samples taken farther apart. There are different types of Kriging techniques, ordinary Kriging is the most common method; however, if there is a spatial trend then this technique is no longer the appropriate to model the spatial variability. There are other alternatives of Kriging, for instance, universal Kriging and simple Kriging, among others. The properly selection of the Kriging technique is focused on the data characteristics. To summarize, the aim of Kriging is to minimize the variance of estimation errors under the constraint of unbiased estimation [17]. The spatial prediction of Kriging does not require that data to be interpolated follow a normal distribution since Kriging describes the best linear unbiased estimator in the sense of least variance. However, if the data follow a normal distribution, then Kriging becomes the best unbiased spatial predictor. Therefore, telecommunications studies include Kriging to realize a highly accurate radio environment map (REM) [18,19,20] or to enrich the training dataset (to produce a large amount of data) for channel modeling, as is reported in [21].

Additionally, in [15], Kriging was employed to estimate shadowing only in indoor scenarios at 869.6 MHz, 1930.2 MHz, 2400 MHz and 2500 MHz without considering O2I non-line-of-sight (NLOS) links. Unlike the links studied in [15], in [6], the authors considered O2I NLOS; nevertheless, they only focused their study on comparing the performance of standard model predictions. To the best of our knowledge, this is the first time such novel modelling for estimating shadowing through Kriging in O2I links at 3.5 GHz for 5G communication systems has been presented.

The findings of this study engage students in wireless telecommunications, professionals in the industry, and readers with new Kriging-applied insights and help them to effectively reduce the time and costs involved in measurements campaigns to achieve efficient radio coverage estimations.

## 2. Methodology and Data Collection

As an optimum combination of carefully measurements, Kriging and simple path loss models were employed to predict complete system coverage performance in two types of O2I NLOS links. The methodology, measurement equipment, scenarios and procedure are described as follows.

### 2.1. Measurements and Data Collection Procedure

In order to represent typical O2I links, received signal-strength measurements were carried out in two universities in Chile with similar scenarios: the engineering campus of Universidad Diego Portales (UDP) in Santiago, and the main campus of Universidad Técnica Federico Santa María (USM) in Valparaíso. This measurement campaign was employed in [6] to research a completely different objective than the one addressed in this study; here we only employ the NLOS O2I samples. In seeking to analyze NLOS links, the samples collected in the measurement campaigns described two types of links reported in Table 1. Table 2 details the scenarios for measurement campaigns and the NLOS O2I samples collected.

In Figure 1, a basic layout of both scenarios is illustrated; for same side (SS) links, the transmitter system was always placed on the sidewalk 0.7 m away from the building wall, and for opposite side (OS) links, it was moved directly across the street from its previous location. Throughout the different transmitter system settings, it was located at a 5 m height with a 60° depression angle to the wall. The transmitter power of the system was 17.8 dBm ($P_{\mathrm{tx}}$) at a 3.5 GHz continuous wave (CW) and it consisted of a vertically polarized directional patch antenna with 10.2 dB gain ($G_{\mathrm{tx}}$), 60° azimuth and elevation half-power beamwidths. The received power was recorded by a narrowband receiver connected to a vertically polarized half-wavelength dipole with a 2.4 dB gain ($G_{\mathrm{rx}}$). The receiver bandwidth of 200 KHz allowed it to capture any frequency dispersion affecting the CW transmission. In all field measurements the received power was at least 20 dB above the receiver noise floor of −123 dB.

For the sampling process the receiver antenna was placed on a computer-controlled rotating arm involving measurements of 6° (or 0.105 rad) angular increments on circles of radius 0.4 m at each receiver placement. For each angular position 25 consecutive power samples were collected, in order to verify consistency and averaging, and to remove residual temporal fades, which did not fluctuate by more than ±0.5 dB due to the narrowband static environment. Then, these consecutive samples were averaged to account for the first power sample value and to continue to the next angular position until 60 possible angular positions were completed in a 360° circle. Finally, 60 received signal power samples were collected at each receive location, as illustrated in Figure 2. Regarding receiver locations, a total of 308 and 108 sample locations were reported for the O2I NLOS USM and UDP scenarios, respectively. 

Considering the angular increments as θ and the circle radius as r in Figure 3 the resulting separation distance is illustrated. In each location, this method yields a circle with λ/2 separation distance between successive antenna positions because 0.04 m ≡ λ/2. According to [22], the spatial average of λ/2 is in accordance with the shadow-fade correlation distance. 

Thus, to average out the small-scale fades, the resulting average power ($\bar{P_{\mathrm{rx}}}$) at specific locations corresponded to the mean of the 60 samples recorded, which was calculated as follows:(1)$$\bar{P_{\mathrm{rx}}}=10\log\left(\frac{1}{60}\sum_{i=1}^{60}10^{P_{{\mathrm{rx}}_{i}}/10}\right),$$
where $P_{{\mathrm{rx}}_{i}}$ is the received power measured in dBm and i is the number of the sample recorded. An overview of the measurement conditions is presented in Table 3. 

### 2.2. Kriging-Based Channel Model Development

The overall methodology for both SS and OS channel modelling links is illustrated in Figure 4 and described as follows. As a first step, from the resulting O2I measurements, SS and OS NLOS links, the path loss L is extracted in dB as a classical link budget:(2)$$L=P_{\mathrm{tx}}+G_{\mathrm{tx}}+G_{\mathrm{rx}}-\bar{P_{\mathrm{rx}}}.$$

To properly choose tuning samples that accurately reflected the characteristics of the larger measurement campaign, the selections suggested by the authors in [15] about the method and dataset size for tuning selection were considered: first, four classification methods were addressed and compared in terms of the mean absolute error (MAE); the results showed the first method as the most accurate with the lowest MAE of 2.16 dB. This method divided the target area of each scenario into representative zones bounded by concentric circles every 5 m from the position of the BS; and second, five approaches were addressed in order to select the least amount of tuning dataset, obtaining the best goodness of fit. The conclusion for this test was to recommend the rate of 60/40 to extract tuning and testing datasets. Considering both suggestions, the measurements were divided into areas defined by rings with radius r=5k in m (where k=1, 2,…, n, and n depends on the maximum separation distance between transmitter and receiver) centered at the position of the BS. Then, a 60/40 rate was extracted from each zone: 60% exclusively used for driving the measurement-based prediction process using Kriging (tuning dataset), and 40% to perform the validation of the estimated path loss at those testing placements (testing dataset). For the model tuning process, the path loss extracted in (2) was defined by two components [22]: median path loss $L_{50}$ and shadowing $L_{s}$ in dB:(3)$$L=L_{50}+L_{s}.$$

The median path loss described how the transmitted signal was attenuated during the path, in terms of the free space path loss $L_{\mathrm{fs}}=20\log_{10}\left(d_{0}\right)+20\log_{10}\left(f\;\left[\mathrm{MHz}\right]\right)-28$ [23], the distance-dependent relation, and specific sources from wall attenuations. In this channel modeling proposal, the median path loss was calculated as:(4)$$L_{50}=L_{\mathrm{fs}}+10n\log_{10}\left(\frac{d}{d_{0}}\right)+e,$$
where $L_{fs}=42.9$ dB, $d_{0}$ is the reference distance of 1 m, n denotes the path loss exponent, d is the Euclidean distance in a three-dimensional space in m, and e is strictly related to attenuation sources such as floor and walls attenuations. The variables n and e are found by fitting a linear equation to the path loss extracted in (2); the linear regression results are shown in Table 4.

In Figure 5 the path loss extracted from (2) is illustrated in blue circles markers for the SS links and in red cross markers for the OS links (where the marker points are already averaged over all 60 samples per received position) along with the median path loss calculated by (4) with the values presented in Table 4 for the SS and OS NLOS O2I links.

For shadowing extraction and continuing with the second component for path loss, the shadowing is extracted from (3) as follows:(5)$$L_{s}=L-L_{50}.$$

The shadowing generation process employs the shadow values previously obtained in (5) to interpolate the known data $L_{s}\left(c_{i}\right)$ in unknown locations $c_{0}$, achieving the accurate shadowing values through the Kriging-aided channel. In other words, from the N shadowing tuning samples at the coordinates $c_{i}$, the dataset vector defined by (6) is extracted, leading Kriging to estimate an unknown shadowing value $L_{s}\left(c_{0}\right)$ at a random location $c_{0}$ from the known samples $L_{s}\left(c_{i}\right)$:(6)$$L_{s}\left(c_{i}\right)={\left(L_{s}\left(c_{1}\right),\;L_{s}\left(c_{2}\right),\;\ldots ,\;L_{s}\left(c_{N}\right)\right)}^{T}.$$

To provide these predictions, Kriging employs the variography to understand and find a pertinent threshold of neighboring samples to the interpolation. In order to summarize the central shadowing tendency, an exponential model function is fitted to the variogram estimator to further visualize the shadowing spatial process. The selection of the exponential function is focused on both theoretical reasons, which highlight the properties that a function selected as a variogram model must fulfil, and practical reasons, for which evidence from multiple studies [15,24,25] demonstrates that the exponential function provides the best fits.

In Figure 6, it is possible to observe that the parametric curve (exponential function, Exp) fits reasonably well over the first three 5 m lags for both SS and OS NLOS links. According to geostatistics [26], this is appropriate for Kriging due to near points, such as shadowing neighboring samples, carry more weight than more distant ones to the unknown shadowing values. Thereby, considering the characteristics of the data, ordinary Kriging is employed to interpolate and then predict shadowing at each grid location in the scenario area. Ordinary Kriging uses a weighted average of the neighboring points to estimate the value of an unobserved point. To guarantee that the estimates are unbiased this Kriging determines the weights under the constraint described below in Algorithm 1. As is illustrated in Figure 4, at the end of the methodology proposed, the estimated path loss is assessed by (3) at the testing locations in order to proceed with the validation of the shadowing measured at those points.

All the supporting information for the Kriging-aided method was analyzed and described in greater depth in [15]; however, an overview is presented in the algorithm below.
**Algorithm 1**. Kriging-aided process.**Designing the path loss:**1:Tune the parameters n and e in (4) from the tuning dataset.2:Extract shadowing from (5).3:Generate shadowing trough the Kriging-aided process: variogram and interpolation.**Modeling the variogram:**4:Calculate the experimental variogram. 5:Summarize the experimental variogram. 6:Fit a parametric curve.7:**if** variogram fits to the first 3 lags **then**
**Kriging interpolation:**8:Create the scenario through a grid.9:**Ordinary Kriging** constraint: $\overset{N}{\underset{i=1}{\sum }}w_{i}=1$10:Calculate the weights $w_{i}$ and the Lagrange multiplier.11:Estimate $L_{s}$ trough the weighted average of the neighboring points.**Estimating the path loss:**12:$L=L_{\mathrm{fs}}+10n\log_{10}\left(\frac{d}{d_{0}}\right)+e+L_{s}$

## 3. Results

The developed model described in Section 2.2 is labeled as *K* with $L=L_{50}+L_{s}$ where $L_{s}$ is extracted from the shadowing grid generated by Kriging. In order to validate the performance and the accuracy of the prediction results of the model, it is evaluated based on the mean error, standard deviation and root mean square error (RMSE) of the predicted path loss values, ${\hat{L}}_{i}$, at the testing measured locations, $t_{N}$, relative to the corresponding testing measured path loss values, $L_{i}$. The mean error (ME), the standard deviation of the error (S) and the RMSE are given by:(7)$$\mathrm{ME}=\frac{1}{t_{N}}\sum_{i=1}^{t_{N}}\left(L_{i}-{\hat{L}}_{i}\right),$$
(8)$$S=\sqrt{\frac{1}{t_{N}-1}\sum_{i=1}^{t_{N}}{\left|\left(L_{i}-{\hat{L}}_{i}\right)-ME\right|}^{2}},$$
(9)$$\mathrm{RMSE}=\sqrt{\frac{1}{t_{N}}\sum_{i=1}^{t_{N}}{\left(L_{i}-{\hat{L}}_{i}\right)}^{2}}.$$

Furthermore, these results are compared with those of the most traditional assumption of $L=L_{50}+L_{s}$, where the shadowing, $L_{s}$, is a random variable with a zero mean and a standard deviation σ extracted from the tuning dataset behavior, labeled as *R*. 

Based on the approach that a typical O2I path loss prediction model considers: the main large-scale propagation loss in line-of-sight (LOS) or NLOS up to the building wall, is a penetration factor that adds wall losses and an indoor path loss term, and that this structure is shared by the most widely used standard models for O2I: COST231 Building Penetration LOS model [7], WINNER+ O2Ia/LOS model [8], ITU-R O2I model [9] and 3GPP 3D-UMi O2I model [10]. The *K* model predictions are also compared with those of the standard models. The path loss estimated trough the standard models is calculated using (7), where each component is in dB and is described in Table 5:(10)$$L_{sm}=L_{1}+L_{2}+L_{3},$$

The path loss components, described in Table 5, were extracted from the setup and results of the O2I NLOS measurement campaign, i.e., the distances of the x, y, and z axes were given by the location of each measurement, the frequency f in Hz ($3.5\times 10^{9}$ Hz) and the room depth w in m by the measurement setup. In each room where the measurements were performed the receiver antenna was placed at three different room depths: around 1 m from the exterior university wall, in the center of the room, and around 1 m from the interior university wall. 

To make a robust analysis to the choice of the 60/40 rate for tuning and testing datasets, a uniform random sampling method was performed 2000 times to estimate the corresponding path loss according to the Kriging-aided model proposed. The average of the 2000 iterations for the mean error, the standard deviation of the error and the RMSE is presented in Table 6 for each link described in Table 2.

In addition, after the 2000 iterations are performed to select a different 60/40 rate from measurements to estimate the path loss, the model-based results are presented in terms of the cumulative distribution function (CDF) of the mean error, the standard deviation of the error and the RMSE in Figure 7, Figure 8 and Figure 9, respectively.

According to Table 6 and Figure 9, in both links, with an RMSE of 2.8 dB, the Kriging-aided model estimates the path loss with a higher level of confidence than the other approaches evaluated. Additionally, this novel technique provides the best fit to the measured data with respect to the standard models presented by Castro et al. in [6]. 

Kriging, as a geostatistical technique, assumes that there is an implied connection between the measured data value at a point and its location in space. Therefore, it was possible to estimate shadowing from the best set of available sample points (tuning dataset) yielding the K model as the most accurate of those exposed in Table 6, where the metrics and the CDFs illustrated in Figure 7, Figure 8 and Figure 9 suggest the efficiency of the proposed model, since, unlike the others, it considers the characteristics of the selected link. Even though the R model tunes the parameters n, e and $L_{s}$ from the tuning dataset and is considered in telecommunication society as the most traditional model, the WINNER+ is quite accurate for the SS link with an RMSE of 4.11 dB, and the COST231, WINNER+ and 3GPP for the OS case with an RMSE of no more than 3.4 dB.

In Figure 7 it is possible to validate the outstanding K model performance. From the 2000 tests that were carried out, it is possible to conclude that, when estimating the path loss with Kriging-aided shadowing, there is a 95% probability that the mean error of the prediction is less than 1 dB and 1.3 dB for SS and OS links, respectively. Regarding Figure 9, the stability and the confidence to predict the path loss more accurately is guaranteed when the K model is used, with a 95% probability that the RMSE is less than 3.1 dB and 3.25 dB for SS and OS links, respectively. 

The applicability of the standard model results presented in Figure 7, Figure 8 and Figure 9 is discussed as follows: the COST231, 3GPP, and WINNER+ models are more accurate for the OS than for SS links, with median errors less than or equal to 1.16 dB and RMSEs of no more than 3.49 dB. The results presented in Table 6 demonstrate the WINNER+ model as the most consistent and the ITU-R as the least accurate compared to the other standard models for both types of links. Regarding SS results and considering the resulting scenario geometry by locating the BS as low as 0.7 m from the building wall, the standard models present higher standard deviations compared to the OS deviations. By taking into account these results and the following two considerations: the standard models were based on measurements at different frequencies: 850 MHz, 1.8 and 1.9 GHz for COST 231; 450 MHz to 6 GHz for WINNER; 2 to 6 GHz for ITU-R; and 0.5 to 100 GHz for 3GPP. Additionally, the formulation of each standard model was originally conceived for cells of up to 1 Km radius. It is important to improve the accuracy of the standard models for a specific application, in this case, for O2I NLOS links at the 3.5 GHz band, one of the candidate frequencies considered for 5G communications.

In [15] it was validated through different indoor approaches that the path loss prediction accuracy was significantly better when Kriging was included as part of the tuning process for frequencies in the ultra-high-frequency (UHF) band. This study validated the potential of this geostatistical technique for O2I scenarios at 3.5 GHz against standard models. Therefore, it is an interesting future line of research to consider other setups to validate and compare Kriging performance against the existing standard models at difference frequencies.

## 4. Conclusions

It was validated that the methodology and the model proposed in this paper for O2I applications such as 5G communications at 3.5 GHz, with a proposed accurate combined path loss and shadowing-aided model, were more accurate and versatile compared to both the conventional linear path loss plus log-normal shadowing model and the existing standard models. 

The results and the methodology proposed in this study will help students in wireless telecommunications, professionals in the industry, and engineers to achieve efficient radio coverage estimation; estimate measurements in situations where the possibility to collect large amounts of data from measurement campaigns is very limited, reducing time and costs in practical campaigns; and to encourage them to perform Kriging-aided channel design, considering its accuracy to predict path loss in O2I NLOS links.

## Figures and Tables

**Figure 1 sensors-21-06716-f001:**
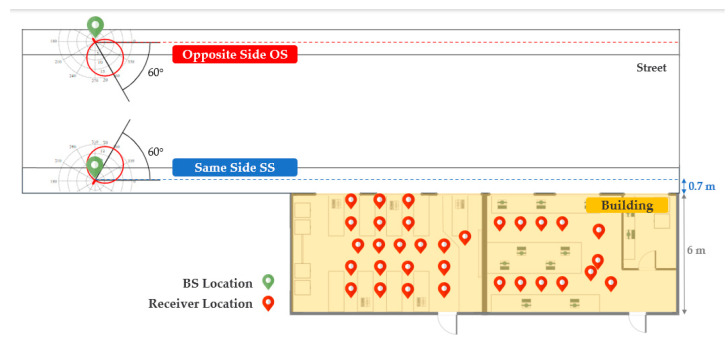
An example of the measurement layout.

**Figure 2 sensors-21-06716-f002:**
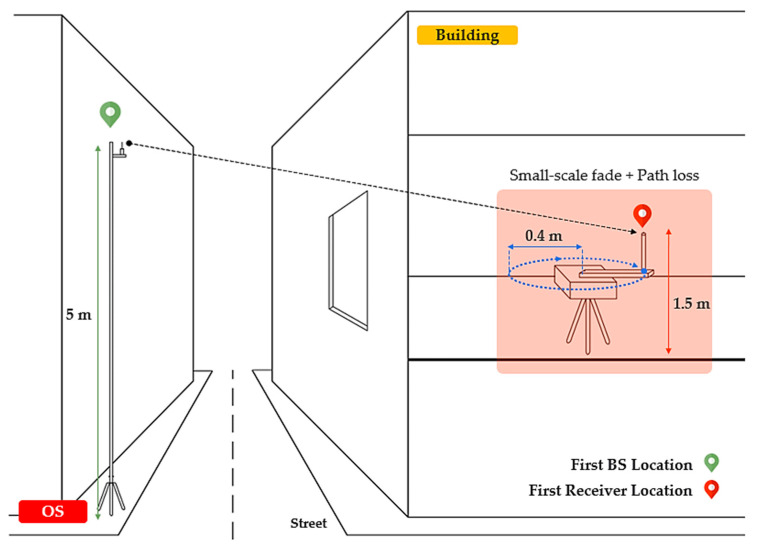
An example of the OS NLOS measurement configuration with the on-axis 0.4 m rotating system.

**Figure 3 sensors-21-06716-f003:**
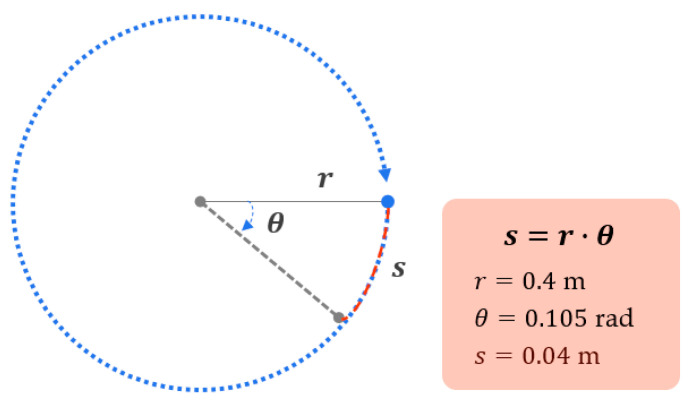
Separation distance according to the angular increment.

**Figure 4 sensors-21-06716-f004:**
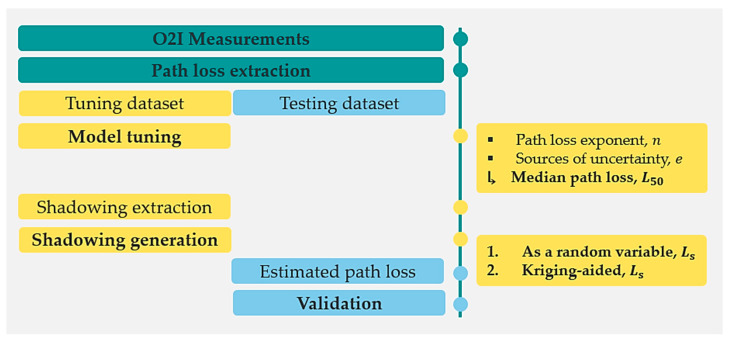
Channel model methodology.

**Figure 5 sensors-21-06716-f005:**
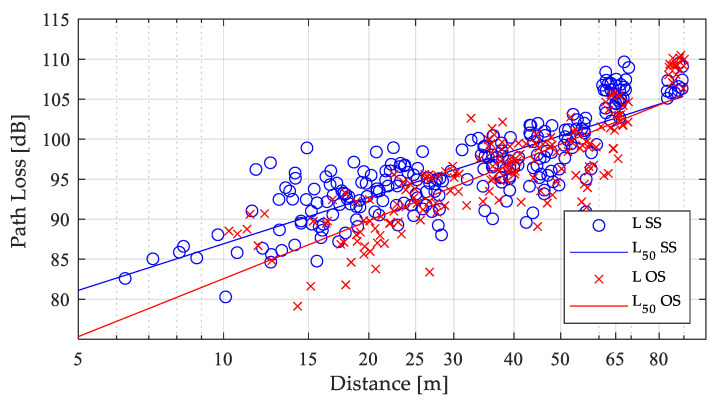
O2I path loss measurements.

**Figure 6 sensors-21-06716-f006:**
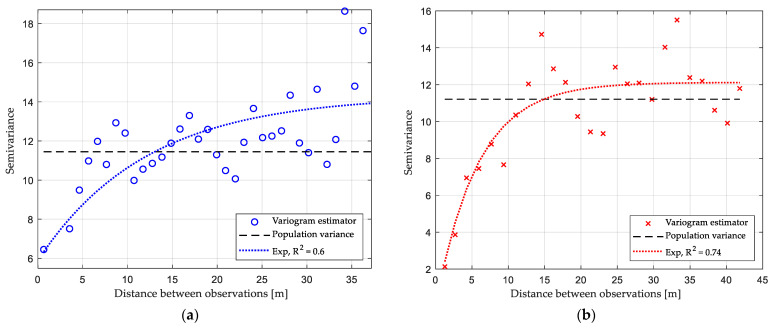
Variography results. (**a**) SS UDP & USM O2I link. (**b**) OS UDP & USM O2I link.

**Figure 7 sensors-21-06716-f007:**
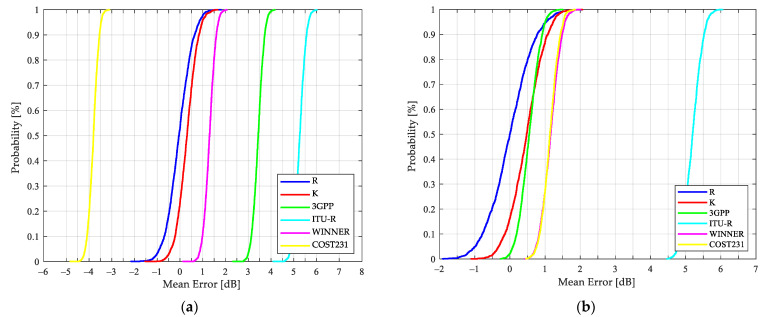
CDFs of mean error for path loss predictions based on different models: (**a**) SS O2I links; (**b**) OS O2I links.

**Figure 8 sensors-21-06716-f008:**
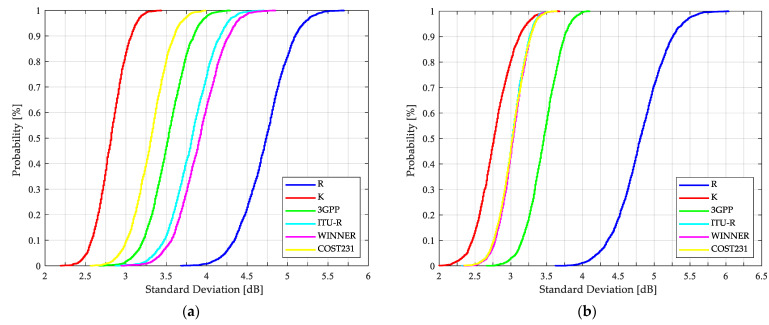
CDFs of standard deviation for path loss predictions based on different models: (**a**) SS O2I links; (**b**) OS O2I links.

**Figure 9 sensors-21-06716-f009:**
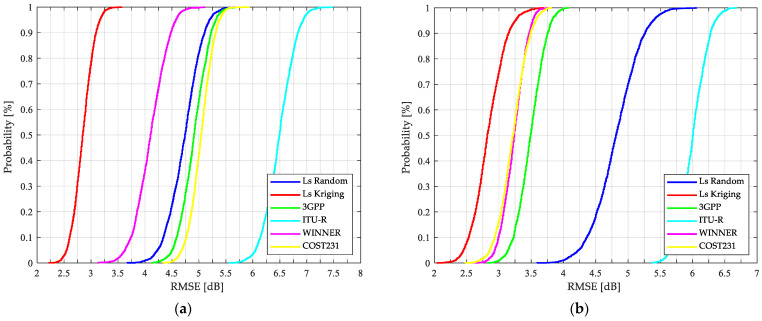
CDFs of RMSE for path loss predictions based on different models: (**a**) SS O2I links; (**b**) OS O2I links.

**Table 1 sensors-21-06716-t001:** Measurement Campaign non-line-of-sight (NLOS) Links.

NLOS Link	Description
Same side (SS)	The transmitter and receiver system share the same street
Opposite side (OS)	The transmitter and receiver system are placed on opposing sides of the street

**Table 2 sensors-21-06716-t002:** Measurement Campaign Scenarios.

Scenario	Outdoor ^1^			Indoor	
	Street Width	Street Length	Surroundings	No. Rooms	Rooms Width
UDP	21 m	89 m	Concrete buildings with different floor heights and a few trees with ≈6 m in height	2	6 m
USM	8.5 m	70 m	Three-story concrete buildings of uniform shape with 6 m^2^ windows	5	6 m

^1^ There is an empty space on the opposite side of the university buildings.

**Table 3 sensors-21-06716-t003:** Measurement Parameters.

Parameter	Description
Type of O2I links	OS NLOS and SS NLOS
Frequency, f	3.5 GHz
$\mathrm{Transmit}\;\mathrm{power},\;P_{\mathrm{tx}}$	17.8 dBm
$\mathrm{Transmit}\;\mathrm{gain},\;G_{\mathrm{tx}}$	10.2 dB
$\mathrm{Receive}\;\mathrm{gain},\;G_{\mathrm{rx}}$	2.4 dB
Receive distance range, d	5–40 m
Receiver noise floor	−123 dBm
Number of spatial positions at each receive location	60
Number of O2I NLOS links at USM	308
Number of O2I NLOS links at UDP	108

**Table 4 sensors-21-06716-t004:** Median Path Loss.

O2I Link	n	e
SS	1.93	24.73
OS	2.41	15.64

**Table 5 sensors-21-06716-t005:** Standard Models.

Standard Model	Path Loss Components ^1^
3GPP	$L_{1}=22\log\left(\sqrt{x^{2}+y^{2}+z^{2}}\right)+28+20\log f$
	$L_{2}=20$
	$L_{3}=\frac{w}{2}$
ITU-R	$L_{1}=22\left(\log\sqrt{x^{2}+{\left(y-w\right)}^{2}+z^{2}}\right)+28+20\log f$
	$L_{2}=14+15\cdot {\left(1-\theta \right)}^{2}$
	$L_{3}=\frac{w}{2}$
WINNER	$L_{1}=22.7\log\left(\sqrt{x^{2}+{\left(y-w\right)}^{2}+z^{2}}\right)+27+20\log f$
	$L_{2}=17.64+14\cdot \left(1-1.8\log f\right)+15\cdot {\left(1-\theta \right)}^{2}$
	$L_{3}=\frac{w}{2}$
COST231	$L_{1}=20\log\left(\sqrt{x^{2}+{\left(y-w\right)}^{2}+z^{2}}+w\right)+32.4+20\log f$
	$L_{2}=7+20\cdot {\left(1-\frac{y-w}{\sqrt{x^{2}+{\left(y-w\right)}^{2}+z^{2}}+w}\right)}^{2}$
	$L_{3}=0.6\cdot \left(w-2\right)\cdot {\left(1-\frac{y-w}{\sqrt{x^{2}+{\left(y-w\right)}^{2}+z^{2}}+w}\right)}^{2}$

^1^ $\theta =\tan^{-1}\left(\sqrt{x^{2}+z^{2}}/y-w\right)$.

**Table 6 sensors-21-06716-t006:** Path Loss Models Accuracy.

O2I Link	Model	Mean Error	Standard Deviation	RMSE
SS	R	−0.04	4.73	4.73
	K	0.30	2.82	2.85
	3GPP	3.43	3.52	4.91
	ITU-R	5.26	3.82	6.49
	WINNER+	1.29	3.91	4.11
	COST231	−3.81	3.31	5.04
OS	R	0.00	4.81	4.81
	K	0.47	2.78	2.84
	3GPP	0.55	3.46	3.49
	ITU-R	5.20	3.02	6.00
	WINNER+	1.16	3.03	3.23
	COST231	1.14	3.02	3.21

## Data Availability

The data presented in this study are available on request from the corresponding author.

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
