# Peer review of "Testing a 5G Communication System: Kriging-Aided O2I Path Loss Modeling Based on 3.5 GHz Measurement Analysis"

_sensors, 2021, doi:10.3390/s21206716_

Round 1

Reviewer 1 Report

Some of my concerns are as follows.

  1. In Section 1, the description of "Kriging is considered the best linear unbiased predictor" is not clear, please define the meaning of best in terms of physical meaning.
  2. Please highlight the main contributions of this work compared with the existing related works.
  3. For the simulation results, e.g., Fig. 6 ~ Fig. 8, are the curves obtained by measurements? It seems the measurements-based curves should not be so smooth. The results are more like model-based results.

Author Response

Dear Reviewer,

I am sending you the responses highlighted in yellow color to your comments to submission Manuscript ID: sensors-1372280. In the first place, I would like to thank you for their kind and thoughtful comments, which indeed improve the quality of the submitted paper.

Reviewer 1 Comments:

In Section 1, the description of "Kriging is considered the best linear unbiased predictor" is not clear, please define the meaning of best in terms of physical meaning.

Author response: We thank and agree with the reviewer comment.

Author action: Following reviewer comment, we updated Section 1 with the following complement:

“The spatial prediction of Kriging does not require that data to be interpolated follows a normal distribution since Kriging describes the best linear unbiased estimator in the sense of least variance. However, if data does follow a normal distribution, then Kriging becomes the best unbiased spatial predictor.”

Please highlight the main contributions of this work compared with the existing related works.

Author response: We thank and agree with the reviewer comment. It is very important to highlight the main contribution compared, specially, to [6] and [11]. Therefore, we mentioned that “… in [11], Kriging was employed to estimate shadowing only in indoor scenarios at 869.6 MHz, 1930.2 MHz, 2400 MHz and 2500 MHz without considering O2I non-line-of-sight (NLOS) links.” However, it is necessary to clarify the main contribution of this study compared to [6].

Author action: We have updated Section 1 in the new version of the manuscript with the following complement:

“Unlike the links studied in [11], in [6], the authors considered O2I NLOS; nevertheless, they only focused their study on comparing the performance of standard model predictions.”

For the simulation results, e.g., Fig. 6 - Fig. 8, are the curves obtained by measurements? It seems the measurements-based curves should not be so smooth. The results are more like model-based results.

Author response: We thank and agree with the reviewer comment, it is necessary to clarify the source of the results for Fig. 7 to Fig. 9, which is measurement-based.

Author action: We have updated the name of Figures 7 to 9 and Section 3 in the new version of the manuscript with the following sentences:

“…to estimate the corresponding path loss according to the Kriging-aided model proposed.”

“…after performed the 2000 iterations to select different 60/40 rate from measurements to estimate the path loss, the model-based results are presented in terms of…”

Figure 1. CDFs of mean error for path loss predictions based on different models: (a) SS O2I links; (b) OS O2I links.”

Figure 2. CDFs of standard deviation for path loss predictions based on different models: (a) SS O2I links; (b) OS O2I links.

Figure 3. CDFs of RMSE for path loss predictions based on different models: (a) SS O2I links; (b) OS O2I links.”

Reviewer 2 Report

The paper proposes a new Outdoor to indoor propagation model for the 3,5GHz band. The authors claim that the method is more accurate than the existing methods and demonstrate it by comparing the proposed method with the other models using a dataset developed for another paper. 

think the topic is very interesting and suitable for publication in Sensors, on the other hand, some topics must be better addressed in a newer version of the manuscript, as will be shown in the next paragraphs.

1) Section 2.1 describes the experimental setup that was established for obtaining the dataset used in the paper. I think that this setup must be better described. For example, it was stated that the transmitter is placed within 0.7m from the building wall, but, in Figure 1, this measure is not drawn and the length of the street is 89 meters. In my opinion, all dimensions must be in the picture and be commented in the text,.

2) A discussion about the applicability of the benchmark models in the proposed scenario may render the paper more interesting. Is the proposed model more accurate only for the proposed scenario or this can be generalized for other frequencies and setups? In what conditions were the models from Table 5 obtained?

3) Each point of the dataset is obtained as a 60 sample average. Why this value is chosen? What is the intersample time and how this value correlates with the channel coherence time?

4) The proposed method clusters the measurements using a 5m from the BS. The authors state that this is the best selection method, but the paper cited as reference was published by the same authors. In my opinion, the conclusions from [11] about the affirmation must better described in the manuscript or an independent source should be used.

5) In the text, It is not clear wether equations (2) to (5) where proposed or they are a well-known model from literature. If it is a literature model, the source should be cited. If it is a proposed model, the calculations or models used to obtain the constant 28 in L_fs should be described.

6) Since the ordinary Kriging theory is not common to every reader, a brief introduction before the presentation of the proposed method may render the paper easier to read.

7) I could not find the description of K and models. I infer that those are the methods proposed by the authors. An explicit definition should be given.

8) I think that a better discussion of the CDF presented in Figure 6 to 8 should be given. In some figures, the benchmark methods seems to perform better than the proposed methods. The reasons and a discussion about the applicability of the proposed method should be provided.

Minor fixes:

Variables from (2) should be mentioned in the text and not only in Table 3

Author Response

Dear Reviewer,

I am sending you the responses to your comments to submission Manuscript ID: sensors-1372280. In the first place, I would like to thank you for your kind and thoughtful comments, which indeed improve the quality of the submitted paper.

Reviewer 3 Report

Good practical work, however, in my point of view, paper lacks measurement equipment description; more specifically, reported RMSEs could be within receiver noise values.

Author Response

(The authors gave the same response as above.)

Reviewer 4 Report

Dear Authors,

Overall, an acceptable manuscript.

However, some points have to be improved:

Note: According to the „Data Availability Statement: The data presented in this study are available on request from the corresponding author.“, an anonymous check of the data and their consistency to the figures and text is NOT possible.

#

Please outline/explain the mentions Models “COST231, WINNER+, ITU-R, and 3GPP” shortly, each. Ie. In if few sentences, that a reader gets the gist of the models and their differences.

Please explain in the introduction (shortly) what is “Kringing” for readers, that are not so familiar with it!

In Table 2, 108 and 308 “Samples” are mentioned. How does this fit to L116- 118 ?

“For the sampling process the receiver antenna was placed on a computer-controlled 116 rotating-arm involving measurements in 6° angular increments on circles of radius 0.4 m, collecting 60 received signal power samples at each receive location, as illustrated in Figure 2.” i.e. 108 or 308 are no multiples of 60.

If 108 and 308 refers to position samples, then mention that explicitly! Is this correct? I.e. there are two typey of samples: 108 and 308 positions (position samples), and each 60 measurement samples (power  samples), referring to an angle?

  1. 233: Table 6.Why is the “Mean Error” e.g. 0.00 and RSM 4.81.,?

Or Mean error for one model 0.55 and for another 1.16 but RSME 3.49 and 3.23 respectively?

I would expect that a larger mean error also corresponds to a larger RMSE.

There you have an RMSE which is “large”, and the mean error is 0.00 .

Please provide formulas for both in order to make this understandable.

Please provide also a formula for the calculated Standard deviation, RSME and Mean error, here..

Is it really standard deviation or rather empirical standard deviation or sample deviation ?

The Conclusion should be extended, including also a interpretation on the different results, i.e. Model “R” hat 0.00 Mean error, but the largest standard deviation and a large RSME.

What are the final model parameters obtained from here? Can you provide them in in similar form as e.g. in Table 5? Or at least n, e, and the Kringing-aided L_s ? An interpretation is also a bit missing.

---

SUGGESTIONS for minor improvements and minor (typo or spelling ) mistakes:

  1. 27: availability. Under…

  1. 75: Plural? “between pathes” ?

  1. 80: What is “REM” ? Please explain

  1. 91. Ther should be text between Headlines 2 and 2.1 .

  1. 102 please expand: “opposing sides of the street”

  1. 124: “In each location, this method yields a circle with ?/2 separation distance between 124 successive antenna positions.” Suggestion: Provide an additional figure to visualize this.

  1. 127 minor ERROR “resulting strength measurement (?rxÌ…Ì…Ì…Ì…)”: P is a power not a (signal) strength. Suggestion: “resulting average power”

(L. 128 to 132: Providing this Formula and the explanation is perfect! Very good!)

  1. 133: instead of “Table 3. Measurement conditions.” Perhaps “Table 3. Measurement parameters.” ?

  1. 137 why so much text in italics?
  2. 137: Better: “the path loss L is extracted in dB”

L.139-140: the “,” at the end of the equation is wrong. As the sentence ends there, a “.” Is mission.

  1. 144 : “To choose proper..” or “To properly choose…”

  1. 153: Equation ends with “.”

  1. 163 “attenuation” without s . What are these sources? Please describe them shortly.

  1. 160 – 167: / Table 4: Please provide Lfs and e seperatly. Or just “e” and mention Lfs in the text.

L161: Lfs is constant, here, isn’t it? Is Lf in dB ? Please provide the value there. Is L_fs = 42,9 dB ?

  1. 219, L. 2221 /Table 5: Please provie the Units of the L1, 2, 3 ; i.e. dB

  1. 221 “coordinates (?, ?, ?) in m,” are these really the coordinated or the distances in x,y,z?
  2. 222 “frequency ? in Hz” Really? Isn’t it in MHz ?????

  1. 226: Table 5: What ist “w” ? a “Weight” or the indoor distance?

  1. 227: What is “Theta” please explain and/or provide a figure.

  1. 231: in this line, it is NOT clear what is “mean error”.

  1. 233: Table 6.Why is the “Mean Error” e.g. 0.00 and RSM 4.81.,?

Or Mean error for one model 0.55 and for another 1.16 but RSME 3.49 and 3.23 respectively?

I would expect that a larger mean error also corresponds to a larger RMSE.

How is it possible to have an RMSE which is “large”, if the mean error is 0.00 ???

Please provide formulas for both in order to make this understandable.

References:

Please check small and capitalization of letters:

Furthermore,

  1. 287: Do you have an DOI for that reference? Is it a book or technical report?
  2. 290: Typo: “GHz” instead of “Ghz”?
  3. 300: “Propagation. In” Is this a chapter of the book? Or is the “.” Wrong?

  1. 319: Please provide a Place,

  1. 325: “M.H. MATLAB® recipes for earth sciences, fourth edition. In MATLAB Recipes for Earth Sciences” ??? The Chapter has the same name as the book?

  1. 319 vs. L 325: “1st ed.;” vs “Fourth Edition”. Please unify.

Author Response

(The authors gave the same response as above.)

Round 2

Reviewer 2 Report

I have carefully analysed the newer version of the manuscript and consider that my previous questions were accurately addressed. In addition I have no further questions and recommend paper acceptance.

Author Response

Dear Reviewer,

I would like to thank you for your kind and attentive comments that undoubtedly improved the paper. Thank you very much for your acceptance.

Kind regards,

Reviewer 4 Report

Many thanks for your detailed and comprehensive improvements.

It’ a nicely written paper!

Two remaining minor SUGGESTIONS (no detailed Answer necessary):

  1. 173: “0.04 m” (the Unit in upright, NOT in italic letters)
  2. 217: “MHz” (the Unit in upright, NOT in italic letters)

Author Response

Dear Reviewer,

I would like to thank you for your kind and attentive comments that undoubtedly improved the paper. 

Following your suggestions, lines 176 and 220 have been modified. 

Kind regards,
